# Age-related differences in stair descent balance control: Are women more prone to falls than men?

**Zuzana Kováčiková** [1]◉*, **Javad Sarvestan** [1]◉, **Erika Zemková** [2]◉

1 Faculty of Physical Culture, Department of Natural Sciences in Kinanthropology, Palacký University Olomouc, Olomouc, Czech Republic, 2 Faculty of Physical Education and Sports, Department of Biological and Medical Sciences, Comenius University in Bratislava, Bratislava, Slovakia

◉ These authors contributed equally to this work.
* kovacikzuz@gmail.com

**Data Availability Statement:** All relevant data are within the manuscript and its Supporting Information files.

**Funding:** The research reported in this manuscript has been supported by the Scientific Grant of the

## Abstract

Stair descent is one of the most common forms of daily locomotion and concurrently one of the most challenging and hazardous daily activities performed by older adults. Thus, sufficient attention should be devoted to this locomotion and to the factors that affect it. This study investigates gender and age-related differences in balance control during and after stair descent on a foam mat. Forty-seven older adults (70% women) and 38 young adults (58% women) performed a descent from one step onto a foam mat. Anteroposterior (AP) and mediolateral (ML) centre of pressure velocity (CoP) and standard deviation of the CoP sway were investigated during stair descent and restabilization. A two-way analysis of variance (ANOVA) revealed the main effects of age for the first 5 s of restabilization. Older women exhibited significantly higher values of CoP sway and velocity in both directions compared to the younger individuals (CoP $SD_{AP5}$, 55%; CoP $SD_{ML5}$, 30%; CoP $V_{AP5}$, 106%; CoP $V_{ML5}$, 75%). Men achieved significantly higher values of CoP sway and velocity only in the AP direction compared to their younger counterparts (CoP $SD_{AP5}$, 50% and CoP $V_{AP5}$, 79%). These findings suggest that with advancing age, men are at higher risk of forward falls, whereas women are at higher risk of forward and sideways falls.

## Introduction

The ability to descend stair once or repeatedly is an essential daily locomotion and concurrently indicator of functional mobility and independence in older adults [1–3]. Compared to other frequent daily activities such as walking or reaching the objects, a stair descent imposes higher neuromuscular and sensorimotor demand for controlling the centre of mass during movement [1,4]. With age-related alterations to the neuromuscular and sensorimotor systems, this transition movement can become progressively more demanding to the extent that stair descent becomes a challenging and hazardous task for many older adults, even for those in apparently good state of health and physical fitness [5,6]. An impaired ability to respond quickly to balance disturbances during stair descent, predisposes older adults to a higher incidence of falls [6].

Czech Science Foundation [GA CR, No. 18-16107Y]. The funding source had no involvement in study design; in the collection; analysis and interpretation of data; in the writing of the report; and in the decision to submit the article for publication.

**Competing interests:** The authors have declared that no competing interests exist.

Stair descent falls in older adults are three times more dangerous than stair ascents, and ten times more dangerous than level ground walking falls in terms of injury severity, occurrence of 'post-fall syndrome', and mortality [6–9]. Falls in older people are typically multifactorial events with risk factors that include mainly female sex, decreased muscle strength, impaired balance control, diseases and others [9,10]. As indicated above, women are more prone to falls than men [11,12]. The most common explanation is that they are generally weaker than men across all age groups [13]. Although women undergo lower relative muscle strength losses over time, its critical level is achieved earlier than in men. This is translated into relatively faster degradation of proprioception, and, consequently of balance control [13–15]. However, findings about balance control degradation in relation to age and gender are contradictory and applicable for static conditions only [16–20]. In addition, findings about static balance control are not directly transferable to dynamic conditions [21,22].

It should be also noted that physical activity and overall physical fitness of older adults could play significant role in stair descent balance control and related falls. It has been observed that older men are more physically active than women, when assessed based on the physical activity guidelines [23,24]. However, they engage more in aerobic activities [25,26]. On the other hand, women participate more in balance and strength exercise programs than men [27–30]. Even though many studies have been conducted on stair descent balance control in older adults [4,31–34], none of them examined men and women separately.

As mentioned above, falls are the result of interactions between many factors, including environmental ones [35]. Stepping down is not always conducted on a firm and stable surface in a real-life situation. The stepping surfaces that people encounter on a daily basis can vary, even during the day. In relation to balance control, unstable or compliant surfaces have received much less attention than hard surfaces, despite the fact that they further challenge the already disturbed somatosensory feedback and balance control in older adults [36–38]. In such circumstances, it is highly possible that even a small balance disturbance during a stair descent could results in falling.

This study was designed to investigate gender-and age-related differences in stair descent balance control and restabilization. We hypothesized worse stair descent balance control and restabilization with advancing age in both women and men. The novelty aspects of this study are as follows: (1) stair descent balance control is evaluated during and immediately after stair descent; (2) a foam mat is used as a stepping surface; (3) age-related changes in balance control are evaluated in men and women separately.

## Materials and methods

### Participants

Older adults were recruited through community service providers, the University of the Third Age at Palacký University Olomouc, personalized invitation letters, and through newspaper advertisement. Physical education students were recruited at Faculty of physical culture. Forty-seven older adults (14 men/33 women) and 38 physical education students (16 men/22 women) without orthopedic, neurological, cardiovascular disorders and/or total joint replacements met eligibility criteria and were enrolled in the study. Their basic descriptive characteristics are reported in Table 1. Physical activity of our participants was screened via interview; this, however, was not an inclusion criterion. Both groups were also screened for falls history. The study was prepared within the project entitled Postural stability and its relationship to the muscle strength of selected muscle groups no. 18-16107Y. Project was approved by the Ethics Committee of the Faculty of Physical Culture, Palacký University Olomouc No.24/2017. The committee evaluated the project and found no contradictions with the principles, procedures

**Table 1. Baseline characteristics of young and older participants.**

| Baseline characteristics | Women | | Men | |
| --- | --- | --- | --- | --- |
| | Young (n = 22) | Older (n = 33) | Young (n = 16) | Older (n = 14) |
| Age (years) | 21.0 ± 1.5 | 68.2 ± 4.9 | 21.9 ± 1.6 | 71.2 ± 7.7 |
| Weight (kg) | 61.6 ± 8.0 | 67.9 ± 8.3 | 73.9 ± 8.3 | 89.0 ± 13.9 |
| Height (cm) | 167.2 ± 5.4 | 163.0 ± 7.2 | 178.4 ± 9.0 | 179.5 ± 6.3 |
| BMI (kg/m²) | 22.0 ± 2.6 | 25.6 ± 3.0 | 23.3 ± 2.4 | 27.6 ± 3.4 |
| Number of falls (n)[‡] | 0 | 5 | 0 | 3 |

*Note*: BMI = body mass index. [‡]Number of falls in the last 3 months before enrolment–single cases in each group.

and international guidelines for medical research involving human subjects. Procedures were in accordance with ethical standards for human experimentation the 1964 Helsinki Declaration (and its later amendments). All participants were informed regarding the study protocol and signed an informed consent form.

## Testing protocol

All participants were evaluated for functional lower limb dominance before balance testing. Limb dominance was determined based on a ball kick test, step up test, and balance recovery test [39]. The limb that was used to kick the ball, descend the stair, or recuperate balance after nudging was considered as the functionally dominant limb.

Participants were barefoot. They started from an upright position on the top of the one-step wooden stair (height = 20.8 cm, width = 46.0 cm, and depth = 38.4 cm) that was placed in front of two force plates positioned side by side (AMTI OR6-5, Advanced Mechanical Technology, Inc., Watertown, MA, USA; sampling frequency 200 Hz) and covered by a foam pad (Airex Balance Pad, Airex AG, Sins, Switzerland; height = 6 cm, width = 50 cm, and depth = 41 cm). A schematic of set-up is shown in Fig 1. Participants were instructed to step down at their preferred speed. Stair descent was initiated with the dominant limb, followed by the nondominant limb onto the same level (step-by-step strategy) with subsequent restabilization. The participants had at disposal practice trial and then the three valid trials were recorded. Based on the initial screening, the participants did not use handrails during stair locomotion in their daily lives; therefore, stair descent was performed without using handrails. However, two assistants stood at each side of the participant and provided security in case of balance loss. The presence of any pain, discomfort and/or clinical signs of musculoskeletal disorder during testing was reason for test interruption. Those who wore glasses were tested wearing them.

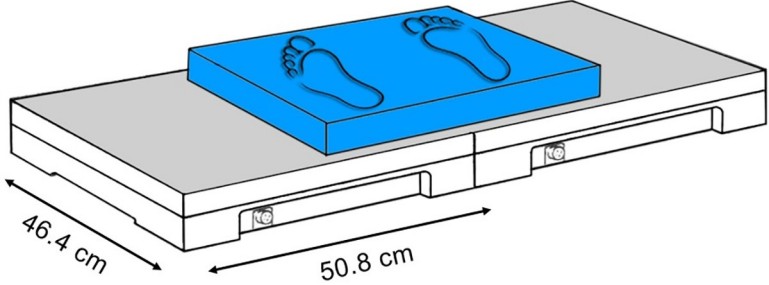

**Fig 1. Schematic of set-up for force platform measurements.**

### Data processing

**Stair descent.** It was defined by the ground contact of the dominant (T0) and nondominant limb (T1). The ground contact of the limbs with the foam mat was defined as the instant when the vertical component of the ground reaction force (vGRF) measured by the force plate exceeded 10 N (Fig 2). A vGRF value of 10 N is often used as a threshold for contact in gait related studies [40,41]. Because the force plates used for the measurement generated a low level of background noise, 10 N was a reasonable threshold value to avoid missing relevant data and was sufficiently high such that noise was not misinterpreted as actual contact. Antero-posterior (AP) and mediolateral (ML) centre of pressure velocity (CoP V) and standard deviation of the CoP sway (CoP SD) were evaluated (CoP $V_{AP}$, CoP $V_{ML}$ and CoP $SD_{AP}$, CoP $SD_{ML}$). In addition, time between the ground contact of the dominant (T0) and nondominant limb (T1) was investigated and considered as stair descent time.

**Restabilization.** It was defined by the ground contact of the nondominant limb with the foam mat (T1) when the vertical component of the ground reaction force (vGRF) measured by the force plate exceeded 10 N (Fig 2). This phase was characterized by the 30 s of quiet standing, however, only the first 5 s were evaluated. Previous studies suggested that the first 5 s after postural perturbations are crucial when assessing balance control in older adults [42,43]. Mean

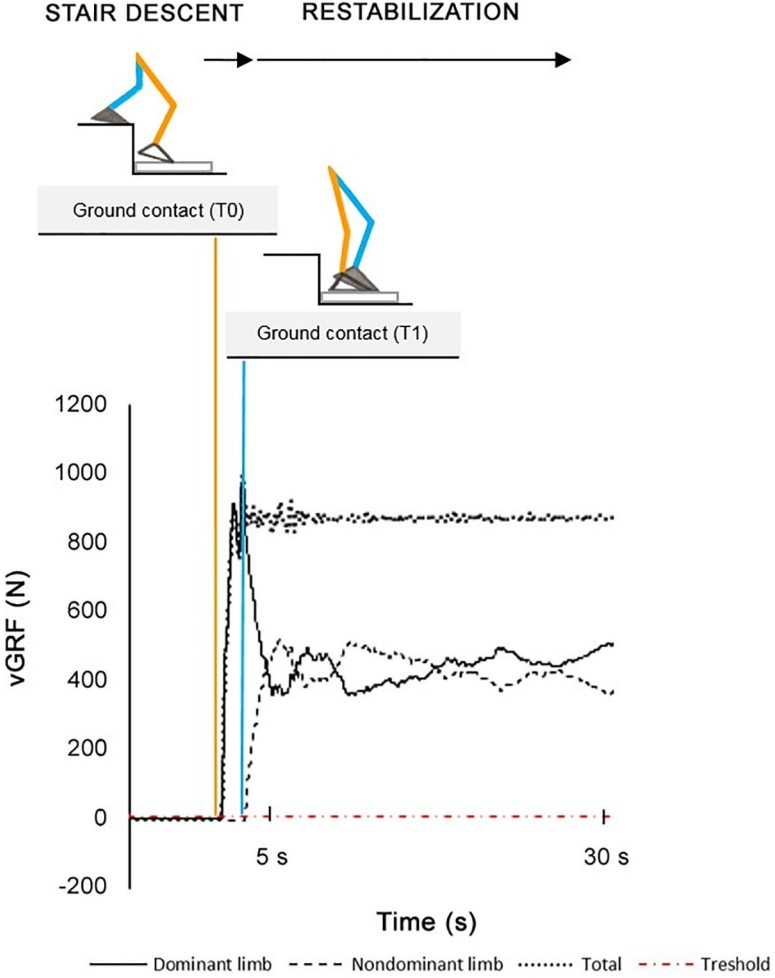

**Fig 2. Definition of stair descent and restabilization based on the vertical ground reaction force.**

CoP V and CoP SD in both directions were evaluated (CoP $V_{AP5}$, CoP $V_{ML5}$ and CoP $SD_{AP5}$, CoP $SD_{ML5}$).

## Data analysis

The CoP coordinates were filtered using a fourth-order low-pass Butterworth filter with a cut-off frequency of 10 Hz. Subsequently, the mean CoP V and CoP SD parameters in both directions were calculated. Data analysis was performed using MATLAB software (v. 2018a, MathWorks, Inc., Natick, MA, USA).

## Statistical analysis

The calculation of the sample size was conducted with $\alpha = 0.05$ (5% chance of type I error) and $1-\beta = 0.80$ (power 80%), and was based on the study of Kim [33], which compared the CoP sway and velocity during a stair descent in young and older adults. A minimum sample size of 13 participants per group was determined to be required in an attempt to minimize the publication bias. Consequently, the sample size employed in our study was regarded as sufficient for detecting significant differences in balance control between groups. The Shapiro-Wilk test was used to test the data distribution. Two-way analysis of variance (ANOVA) was applied in order to assess the effects of gender and age (2 gender groups [males and females] × 2 age groups [young and older]) on the dependent variables. If significant main effect and/or interaction was found, Tukey's post-hoc test was performed. Statistical analysis was performed using Statistical Package for the Social Sciences (SPSS) for Windows, version 25 (SPSS, IBM Corporation, Armonk, NY, USA). The significance level was set at $p < 0.05$.

## Results

The results of two-way ANOVA indicated the significant main effects of age for the first 5 s of restabilization only. The same analysis revealed no significant effects of gender, and interactive effects between age and gender for any dependent variable (Table 2).

**Table 2. Stair descent balance control and restabilization in women and men, by age groups.**

| | Women | | Men | | 2-way ANOVA (p) | | |
| --- | --- | --- | --- | --- | --- | --- | --- |
| **Stair descent** | **Young (n = 22)** | **Older (n = 33)** | **Young (n = 16)** | **Older (n = 14)** | **A** | **G** | **G x A** |
| Descent time (s) | 0.69 (0.608–0.860) | 0.74 (0.607–0.883) | 0.71 (0.64–0.83) | 0.70 (0.66–0.79) | 0.759 | 0.531 | 0.310 |
| CoP $V_{AP}$ (mm.s$^{-1}$) | 291.4 (218.0–481.2) | 366.2 (278.5–792.1) | 332.6 (224.7–779.7) | 393.0 (325.9–637.9) | 0.338 | 0.629 | 0.162 |
| CoP $V_{ML}$ (mm.s$^{-1}$) | 251.3 (138.8–524.9) | 352.9 (241.2–793.3) | 349.1 (164.4–765.1) | 316.0 (225.5–553.1) | 0.252 | 0.713 | 0.382 |
| CoP $SD_{AP}$ (mm) | 30.3 (22.9–35.4) | 31.0 (24.9–39.1) | 29.9 (25.7–43.6) | 33.4 (25.4–37.9) | 0.587 | 0.990 | 0.248 |
| CoP $SD_{ML}$ (mm) | 10.0 (7.4–23.2) | 18.8 (11.5–33.9) | 17.7 (9.6–31.5) | 17.9 (9.2–25.5) | 0.610 | 0.900 | 0.409 |
| **Restabilization** | | | | | | | |
| CoP $V_{AP\,5}$ (mm.s$^{-1}$) | 30.1 (28.1–35.4) | 71.5 (54.2–80.2)*** | 34.6 (27.8–40.2) | 61.5 (43.5–67.4)** | **<0.001** | 0.299 | 0.221 |
| CoP $V_{ML\,5}$ (mm.s$^{-1}$) | 43.6 (37.9–50.2) | 70.9 (61.0–90.6)*** | 52.0 (49.0–60.3) | 67.6 (51.2–86.9) | **<0.001** | 0.594 | 0.061 |
| CoP $SD_{AP\,5}$ (mm) | 9.9 (9.0–11.1) | 16.3 (13.4–19.0)*** | 10.5 (8.9–12.3) | 15.5 (11.6–18.2)** | **<0.001** | 0.734 | 0.722 |
| CoP $SD_{ML\,5}$ (mm) | 16.5 (14.1–20.3) | 22.9 (17.8–26.1)** | 18.2 (16.6–21.1) | 23.3 (18.1–24.4) | **0.003** | 0.782 | 0.399 |

*Note*: CoP $V_{AP}$ = mean CoP velocity in anteroposterior direction; CoP $V_{ML}$ = mean CoP velocity in mediolateral direction; CoP $SD_{AP}$ = the standard deviation of the anteroposterior CoP sway; CoP $SD_{ML}$ = the standard deviation of the mediolateral CoP sway; CoP $V_{AP\,5}$ = mean CoP velocity in the anteroposterior direction during the first 5 s of restabilization; CoP $V_{ML\,5}$ = mean CoP velocity in the mediolateral direction during the first 5 s of restabilization; CoP $SD_{AP\,5}$ = the standard deviation of the anteroposterior CoP sway during the first 5 s of restabilization; CoP $SD_{ML\,5}$ = the standard deviation of the mediolateral CoP sway during the first 5 s of restabilization.; G = gender effect; A = age effect; G × A = gender × age interaction. *Significantly different than younger counterparts ($p < 0.05$).

**Significantly different than younger counterparts ($p < 0.01$).

***Significantly different than younger counterparts ($p < 0.001$). Data are presented as median and interquartile range.

Older women achieved significantly higher values of CoP V and CoP SD in both directions (CoP $V_{AP5}$, 106%, p<0.001; CoP $V_{ML5}$, 75%, p<0.001; CoP $SD_{AP5}$, 55%, p<0.001; CoP $SD_{ML5}$, 30%, p = 0.009) and were considered as less stable compared to young women. Older men exhibited higher values of CoP V and CoP SD direction than their younger counterparts, however, in the AP direction only (CoP $V_{AP5}$, 79%, p = 0.004; CoP $SD_{AP5}$, 50%, p = 0.003). Results are summarized in Table 2.

## Discussion

This study investigated gender-and age-related differences in stair descent balance control and restabilization. Only the first 5 s of restabilization differentiated significantly between young and older participants. Women presented higher values of CoP sway and velocity in both directions with age, while men achieved higher values of CoP sway and velocity only in the AP direction compared to their younger counterparts.

We found two studies that looked at the effect of aging on balance recovery after taking a step [44,45]. In both studies, lesser stability after step execution and slower balance recovery to a stable state were observed in older adults compared to younger participants. However, we cannot compare our results with those obtained in the above-mentioned studies since step-over-step stair descent on hard surface was evaluated and mixed gender groups were compared.

The CoP velocity and sway measures have been previously used to evaluate age-related changes in balance control [33,46,47] with higher values attributed to worse balance [46,48]. The CoP velocity relies primarily on greater sensory integration from the visual and vestibular systems, whereas sway is controlled principally by the somatosensory system [46,49]. The larger values of CoP sway, along with higher CoP velocity observed in our study could indicate deficits in the sensory and somatosensory system resulted in decreased steadiness after stair descent [46,50,51].

In our study, we used a foam mat as a stepping surface. Stepping on a compliant surface increases stress applied to the sensory system. The sensorimotor loading and integration between the systems are critical in determining body positioning in space [52]. Deficits in the integration of the sensory systems could increase the displacement of the center of mass outside of the base of support and induce fall [52]. Stepping down is performed not only during stair descent, but also in many other everyday activities such as stepping down from a curb, alighting from public transport, or walking downhill. To perform these activities in a safe manner can be threatened by various ground surface changes. A compliant surface such as carpet, grass, snow, or a sand reduces ability of kinesthetic system to accurately detect body orientation with respect to the ground surface. In addition, it induces a mechanical perturbation which is caused by the compression of an extremely compliant viscoelastic surface during stepping [53]. In such circumstances, stepping down activity can much easily lead to falls with less or more serious health consequences.

Multifactorial origin of falls is well-known. The higher the number of risk factors, the more likely it is that a fall will occur, with about an 8% falls rate with no risk factors and then a 15–20% increased risk for each additional risk factor [9]. Despite the fact that our older men and women were physically active, combination of factors such as advancing age, decreased balance control in the first 5 s of restabilization, compliant ground surface, gender and previous history of falls could increase the risk of falling.

Compared to younger counterparts, older women were less stable in both the AP and ML directions, whereas older men were less stable only in the AP direction. It is generally known that older adults are less efficient at using ankle strategy, which is key to AP balance control

[45]. Reduced AP balance control predisposes older adults to a higher incidence of stair descent falls, which generally occur in the anterior direction [1,54,55]. In contrast, relatively less attention is typically paid to ML balance control in relation to stair descent locomotion despite the fact that the ML sway is a predictive variable for sideways falls [33]. Sideways falls increase the risk of hip fracture six-fold as compared to forward or backward falls [56], and are associated with increasing disability, morbidity, and mortality [57–60]. Most falls in older adults occur when the leading limb moves onto the lower level, and the trailing limb begins to bend at the knee and hip [61]. In other words, they occur during the stair descent. It was therefore quite unexpected to find significant age-related differences in balance control in the restabilization, but not during the stair descent. Accordingly, any limitation in balance control in restabilization may negatively affect the subsequent step, and more generally, the continuous step-by-step stair descent.

Balance control deterioration may be slowed down by appropriate interventions, which should be aimed at improving the safety of stair descent locomotion in older adults. Even though the older adults in our study could be considered as physically active, their physical activity was predominantly aerobic (e.g. walking, Nordic walking, hiking, swimming, cycling) and therefore, had no or minimal impact on balance control. A training load of aerobic character (persistent effort of low intensity) is not a sufficient stimulus for the development of balance abilities. Training specificity is widely established as an integral aspect affecting training responses. Therefore, even our relatively healthy and physically active older adults may be at risk of falls. According to the World Health Organization, older adults should perform physical activity to enhance balance and prevent falls three or more days per week [62]. To achieve the greatest improvements, training interventions should emphasize simulating not only movement patterns, but also other important factors, such as environmental conditions.

Older adults were less effective in controlling balance disturbances in the first 5 s of restabilization than their younger counterparts. Older women exhibited significantly worse balance control in the AP and ML directions than their younger counterparts, whereas men were significantly less stable only in the AP direction with advancing age. This could indicate that men are at a higher risk of forward falls with advancing age, whereas women are at higher risk of falls in both directions. Evaluation of CoP velocity and sway after stair descent on a compliant surface provides a simple way of measuring age-related changes in balance control under realistic conditions and could prove useful when screening for balance disorders, not only in those prone to falls. Some limitations of this study have to be noted. The different number of young and older participants and the overall small number of men included can be considered as the first limitation. Approximately the same number of women and men were recruited for each age category, but only a small number of men met the strict inclusion criteria. Moreover, women were more willing to participate in this study. A second limitation of this study was that the single step was evaluated only. The next limitation can be considered the absence of fallers in the group of young adults.

## Supporting information

**S1 Data.**
(XLSX)

## Acknowledgments

The authors thank Dr. Juraj Pecho for his help with Figures.

## Author Contributions

**Conceptualization:** Zuzana Kováčiková, Javad Sarvestan, Erika Zemková.

**Data curation:** Zuzana Kováčiková, Javad Sarvestan.

**Formal analysis:** Zuzana Kováčiková, Javad Sarvestan.

**Investigation:** Zuzana Kováčiková, Javad Sarvestan.

**Methodology:** Zuzana Kováčiková, Javad Sarvestan.

**Project administration:** Zuzana Kováčiková.

**Supervision:** Zuzana Kováčiková.

**Writing – original draft:** Zuzana Kováčiková, Javad Sarvestan, Erika Zemková.

**Writing – review & editing:** Zuzana Kováčiková, Javad Sarvestan, Erika Zemková.

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
