## [Decision Letter · Decision Letter 0]

6 Nov 2020

PONE-D-20-27933

Age-related differences in stair descent balance control: are women more prone to falls than men?

PLOS ONE

Dear Dr. Kovacikova,

Thank you for submitting your manuscript to PLOS ONE. After careful consideration, we feel that it has merit but does not fully meet PLOS ONE’s publication criteria as it currently stands. Therefore, we invite you to submit a revised version of the manuscript that addresses the points raised during the review process.

Both reviewers suggested major revisions for your manuscript. I agree with them, especially related to methodological issues (e.g., description of stair descent) and statistical analysis. Please address adequately the question about statistical analysis. I think that the reviewer is right. If you disagree, you need a good explanation for the analysis employed.

We look forward to receiving your revised manuscript.

Kind regards,

Fabio A. Barbieri, PhD

Academic Editor

PLOS ONE

Journal Requirements:

Reviewers' comments:

Reviewer's Responses to Questions

**Comments to the Author**

1. Is the manuscript technically sound, and do the data support the conclusions?

Reviewer #1: Yes

Reviewer #2: Yes

2. Has the statistical analysis been performed appropriately and rigorously? 

Reviewer #1: Yes

Reviewer #2: Yes

3. Have the authors made all data underlying the findings in their manuscript fully available?

Reviewer #1: Yes

Reviewer #2: Yes

4. Is the manuscript presented in an intelligible fashion and written in standard English?

Reviewer #1: Yes

Reviewer #2: Yes

5. Review Comments to the Author

Reviewer #1: General comments

This study aimed to investigate age-related differences in balance control during and after stair descent in older men and older women.

The subject addressed is relevant and has practical applicability. However, some aspects need to be improved and clarified.

Specific comments

Abstract

1 – The objective presented in the abstract does not fully match with the title of the study, because it was not informed in the objective that men and women would be separately analyzed.

2 – P.2, lines 34-35: this sentence is too vague. Please, specify what were the age-related differences.

3 – It is not clear what the percentages showed are representing (e.g. mean, range, CoP sway, CoP velocity, AP direction, ML direction…). Please, clarify in the text.

4 – I suggest including a keyword related to postural control.

Introduction

5 – It was stated in the aim of the study that gender- and age-related differences would be investigated. However, men and women were not compared. It is necessary to rewrite the aim accordingly.

Materials and Methods

6 – In Table 1, it is not clear what is the number of falls presented. Are they the mean of falls of all older women and men included in the study? Were all the older individuals fallers?

7 – Do you have information about the type of falls they had suffered (e.g. sideways, forward, backward)? If yes, this could be an interesting information to discuss the results found.

8 – I am not sure whether the force plates were positioned side by side or one in front of each other. I suggest including a figure to show this setting.

9 – What was the size of each force plate? Was there a foam pad for each force plate? Did the foam mat cover exactly the total area of each force plate? How was possible to know when the participant stepped down with the entire foot onto each force plate? Please, include this information in the Methods section.

10 – It was informed that “only the first 5 s were evaluated”. When did this time start? Did you evaluate 5 s for each force plate and each limb?

11 – How was the stair descent time determined? Please, include this information in the text.

12 – Please, make it clearer in the Methods how were the phases named and determined. The terms “stair descent phase”, “restabilization phase”, “stair descent”, “during stair descent”, and “after stair descent” were utilized, but it would be more clear if only one term would be utilized for each phase, and whether a clear description on how each phase was determined was included.

Results

13 – It is not clear whether the stair descent phase data are from the dominant limb and the restabilization phase are from the non-dominant limb, or whether one of these phases are from both limbs. Please, clarify.

Discussion

14 – In the Introduction, the authors cited that men and women use to practice different types of exercise. What were the exercises and the weekly number of sessions practiced by the participants of the study? Could the differences found for men and women be related to the exercises they have been regularly practicing? I suggest including a paragraph about this in the discussion.

Reviewer #2: General comments:

This study examined the age-related difference in balance control during and after a stair descent on a compliant surface. The strength of this paper was to test the sex-difference of the balance control after stair descent on a compliant surface in two age groups. I will make some recommendations for major and minor revisions to improve the clarity of the manuscript as follows.

1. In the title, “Age-related differences” let leaders expect to see the test of the age difference, and the hypothesis also includes the effect of advancing age in both women and men. However, in the statistical analysis, one-way ANOVA was used. To interpret the data as a function of both age and sex, two-way ANOVA would be appropriate. For example, you may have an interaction of sex by age in any of your dependent variables. Otherwise, if the purpose is only to look at the differences between age groups within a gender, then the title and/or hypothesis may need to be updated for clarity.

2. “One-step wooden staircase” sounds more like a curb instead of a staircase. Did the subject stand on the wooden step and step down? Is there any rationale that it is described as stair descent instead of stepping down a curb?

3. For clarification, please describe clearly the stair descent phase and the restabilization phase. Is the stair descent phase from first foot contact on the ground to the next foot contact on the ground? Is the restabilization phase includes 5 seconds from the moment when both force plates over 10N?

4. Line 235: What do other important factors mean?

Minor revisions

1. A wording for “older adults” supposed to be consistent for all of the manuscript, but the authors used ‘older adults’ mixed with ‘elderly adults’ [line 49, 76, 186].

2. Table1: How did you get the number of falls in older adults? Was it a retrospective question? Why does no data for young adults?

3. Table2 has a different format for each cell. It should be described as standard deviation or range consistently.

6. PLOS authors have the option to publish the peer review history of their article (what does this mean?). If published, this will include your full peer review and any attached files.

Reviewer #1: No

Reviewer #2: No

---

## [Author Response · Author response to Decision Letter 0]

11 Nov 2020

We would like to thank Editor and the Reviewers for sparing the time to write us so many detailed and useful comments and recommendations. All changes in the manuscript are highlighted. 

Below you will find our point-by-point statements to the reviewers’ comments. Changes are highlighted in yellow for reviewer #1 and in green for reviewer #2. Other associated changes are highlighted in purple. 

Yours sincerely

The authors

Reviewer #1

Specific comments

Abstract

1 – The objective presented in the abstract does not fully match with the title of the study, because it was not informed in the objective that men and women would be separately analyzed.

Statement: We thank Reviewer for his/her comment. Based on the recommendation (Reviewer 2), a two-way analysis of variance (ANOVA) was applied in order to assess the effects of gender and age on the dependent variables. Men and women were analyzed together, and results were changed according to the new findings. Lines 29-30.

2 – P.2, lines 34-35: this sentence is too vague. Please, specify what were the age-related differences.

Statement: Done. Mentioned sentence was rewritten (lines 33-35).

3 – It is not clear what the percentages showed are representing (e.g. mean, range, CoP sway, CoP velocity, AP direction, ML direction…). Please, clarify in the text.

Statement: Done. Required information was added. Lines 36 and 38.

4 – I suggest including a keyword related to postural control.

Statement: Done. Balance control and centre of pressure were added. Line 41.

Introduction

5 – It was stated in the aim of the study that gender- and age-related differences would be investigated. However, men and women were not compared. It is necessary to rewrite the aim accordingly.

Statement: We thank Reviewer for his/her comment. Data were analyzed using by 2-way ANOVA (2 gender groups [males and females] × 2 age groups [young and older]) instead of 1-way ANOVA and results were rewritten according to the new findings. We believe that our aim is ok now.

Material and Methods

6 – In Table 1, it is not clear what is the number of falls presented. Are they the mean of falls of all older women and men included in the study? Were all the older individuals fallers?

Statement: Young participants and older adults were screened for falls history. They were asked the following questions: "Did you fall in the last three months?" and "What was the reason of your fall/falls?". Number of falls (n) is presented in Table 1 – single cases in each group. Not all older adults were fallers. Lines 96 and 105.

7 – Do you have information about the type of falls they had suffered (e.g. sideways, forward, backward)? If yes, this could be an interesting information to discuss the results found.

Statement: This is an interesting point. The medical and clinical status of our participants was checked by anamnestic questionnaire and interview. However, the number of falls in the last three months before study enrollment and their reasons were evaluated only. The falling direction was not evaluated. 

8 – I am not sure whether the force plates were positioned side by side or one in front of each other. I suggest including a figure to show this setting.

Statement: Force plates were positioned side by side and covered by one foam pad. This information was added to the Testing protocol (line 114). We absolutely agree with Reviewer that adding a photo could make it easier for readers to understand. Figure 1 was added.

9 – What was the size of each force plate? Was there a foam pad for each force plate? Did the foam mat cover exactly the total area of each force plate? How was possible to know when the participant stepped down with the entire foot onto each force plate? Please, include this information in the Methods section.

Statement: We thank Reviewer for his/ her valuable suggestions. The size of one force plate was 50.8 x 46.4 cm. One foam mat was used. No, total area of each force plate was not covered (based on our pilot measurements, it was not necessary). Participants were informed about the set-up for force platform measurements and the data collection. They were also instructed about the foot placement. These instructions did not adversely affect the movement pattern and we were able to obtain data under the most realistic conditions.

We have implemented the Reviewer’s very useful recommendation into Methods section. Figure 1 was added for better understanding (lines 125-126).

10 – It was informed that “only the first 5 s were evaluated”. When did this time start? Did you evaluate 5 s for each force plate and each limb?

Statement: The onset of the restabilization was defined as the contact of the nondominant limb (T1) with the ground. Time started when nondominant/trailing limb touched the foam mat and the vertical component of the ground reaction force (vGRF) measured by the force plate exceeded 10 N.Both force plates were synchronized. Plates were calibrated before each trial and with foam mat on the top. Restabilization was characterized by bipedal stance with each foot placed on one force plate. The data for each limb/plate were then processed and averaged. Generally, we obtained one output from both plates. Amti force plates are very sensitive and are able to detect data correctly even if the ground contact was performed very close to the force plate edge. Foot placement was checked by examiner and assistants.

Data processing section was rewritten, and Figure 2 was added.

11 – How was the stair descent time determined? Please, include this information in the text.

Statement: Done. This information was added (lines 139-141).

12 – Please, make it clearer in the Methods how were the phases named and determined. The terms “stair descent phase”, “restabilization phase”, “stair descent”, “during stair descent”, and “after stair descent” were utilized, but it would be more clear if only one term would be utilized for each phase, and whether a clear description on how each phase was determined was included.

Statement: We thank Reviewer for his/her valuable recommendation. Whole manuscript was checked for mentioned phrases. We decided to use only STAIR DESCENT and RESTABILIZATION and to minimize other phrases as much as possible. Both phases are defined (see Methods section – Data processing). 

Results

13 – It is not clear whether the stair descent phase data are from the dominant limb and the restabilization phase are from the non-dominant limb, or whether one of these phases are from both limbs. Please, clarify.

Statement: Dear Reviewer, results were rewritten according to the new findings (statistical analysis). Methods section was rewritten, and Figures 1 and 2 were added for better understanding. Very easy explanation is that the stair descent was defined by the contact of dominant and nondominant limb (single limb support phase) and restabilization was characterized by bipedal stance. The aim was to obtain data under realistic conditions (no special instructions, no trunk corrections, no descent pattern corrections…). 

Discussion

14 – In the Introduction, the authors cited that men and women use to practice different types of exercise. What were the exercises and the weekly number of sessions practiced by the participants of the study? Could the differences found for men and women be related to the exercises they have been regularly practicing? I suggest including a paragraph about this in the discussion.

Statement: Young participants were PE students participating on aerobic (40%), intermittent (34%) and balance (26%) activities. All older adults were physically active, participating in organized physical activities and sports for at least two 60 min sessions per week, and walking for more than 2 h per day. Their physical activity has not been analyzed in detail. Their physical activity was aerobic (walking, cycling, Nordic walking, swimming etc.). Based on our experience, older adults are willing to participate on balance interventions. However, these programs are conducted predominantly as a part of scientific studies. Within our scientific project, older adults underwent 8-weeks balance interventions with higher number of women than men in each group. On the other hand, men more often continued to exercise in the home environment than women.

We thank Reviewer for his/her useful recommendation. Paragraph dealing with physical activity was added (lines 244 – 251).

Reviewer#2:

General comments:

1. In the title, “Age-related differences” let leaders expect to see the test of the age difference, and the hypothesis also includes the effect of advancing age in both women and men. However, in the statistical analysis, one-way ANOVA was used. To interpret the data as a function of both age and sex, two-way ANOVA would be appropriate. For example, you may have an interaction of sex by age in any of your dependent variables. Otherwise, if the purpose is only to look at the differences between age groups within a gender, then the title and/or hypothesis may need to be updated for clarity.

Statement: We thank Reviewer for this valuable comment. Data were analyzed using by two-way ANOVA instead of one-way ANOVA (Kruskal-Wallis). Results were rewritten according to the new findings. 

2. “One-step wooden staircase” sounds more like a curb instead of a staircase. Did the subject stand on the wooden step and step down? Is there any rationale that it is described as stair descent instead of stepping down a curb?

Statement: We thank Reviewer for his/her comment. If we understand it correctly, discrepancy is in using word stair. Both, stair descent and also stepping down a curb are using when single step is evaluated. We preferred phrase stair descent also because rehabilitation companies offer wooden stair instead of wooden curb. Staircase was replaced by stair.

Lebleu, J., Mahaudens, P., Pitance, L., Roclat, A., Briffaut, J. B., Detrembleur, C., & Hidalgo, B. (2018). Effects of ankle dorsiflexion limitation on lower limb kinematic patterns during a forward step-down test: A reliability and comparative study. Journal of back and musculoskeletal rehabilitation, 31(6), 1085-1096.

Li, Z. Y., & Chou, C. (2014). The effect of cane length and step height on muscle strength and body balance of elderly people in a stairway environment. Journal of physiological anthropology, 33(1), 36.

Gao, B., Cordova, M. L., & Zheng, N. N. (2012). Three-dimensional joint kinematics of ACL-deficient and ACL-reconstructed knees during stair ascent and descent. Human movement science, 31(1), 222-235.

3. For clarification, please describe clearly the stair descent phase and the restabilization phase. Is the stair descent phase from first foot contact on the ground to the next foot contact on the ground? Is the restabilization phase includes 5 seconds from the moment when both force plates over 10N?

Statement: Yes, the stair descent is defined by the ground contact of the dominant/leading (T0) and nondominant/trailing limb (T1). The ground contact of the limbs with the foam mat was defined as the instant when the vertical component of the ground reaction force (vGRF) measured by the force plate exceeded 10 N. The onset of the restabilization was defined as the ground contact of the nondominant limb (T1) when the vertical component of the ground reaction force (vGRF) measured by the force plate exceeded 10 N. Figure 1 was added for better understanding. 

 Data processing section was rewritten.

4. Line 235: What do other important factors mean?

Statement: We thank Reviewer for his/her suggestion. We added " such as environmental conditions" at the end of the mentioned sentence (line 254).

Minor revisions

1. A wording for “older adults” supposed to be consistent for all of the manuscript, but the authors used ‘older adults’ mixed with ‘elderly adults’ [line 49, 76, 186].

Statement: We thank Reviewer for his/her valuable comment. Phrase ‘elderly adults’ was replaced by ‘older adults’ through the whole manuscript as suggested. 

2. Table1: How did you get the number of falls in older adults? Was it a retrospective question? Why does no data for young adults?

Statement: The number of falls was obtained from participants through interview. Yes, it was a retrospective question (Q1: Did you fall in the last three months? Q2: What was the reason of your fall/falls?). Physical activity was the only reason for falls in older adults – single fall incidents. Young individuals have not reported falls in the last three months before enrollment.

Table 1 - We used zero (0) instead of symbol ˈ-ˈ for better understanding. 

3. Table2 has a different format for each cell. It should be described as standard deviation or range consistently.

Statement: We change it as suggested. Data in Table 2 are presented as median and interquartile range.

---

## [Decision Letter · Decision Letter 1]

15 Dec 2020

PONE-D-20-27933R1

Age-related differences in stair descent balance control: are women more prone to falls than men?

PLOS ONE

Dear Dr. Kovacikova,

Thank you for submitting your manuscript to PLOS ONE. After careful consideration, we feel that it has merit but does not fully meet PLOS ONE’s publication criteria as it currently stands. Therefore, we invite you to submit a revised version of the manuscript that addresses the points raised during the review process.

Dear authors,

One reviewer accepted the manuscript and another one suggested minor revisions. I already have read the paper. It is very interesting, but I have little suggestions to improve it before acceptance.

1) Please improve the rationale of the abstract and remove it. For example, why are more readily seen differences between age and gender under unstable conditions?

2) Please double check the writing on line 64 and 222.

3) Please change hypothesize to hypothesized. - line 82

4)  I have two possible limitations of the study. Please addressed them. 

I) there are fallers in the older adults sample, but not in young. This should be mentioned.

II) the number of young and older participants (especially for women) is many different. This should be mentioned too.

We look forward to receiving your revised manuscript.

Kind regards,

Fabio A. Barbieri, PhD

Academic Editor

PLOS ONE

Reviewers' comments:

Reviewer's Responses to Questions

**Comments to the Author**

1. If the authors have adequately addressed your comments raised in a previous round of review and you feel that this manuscript is now acceptable for publication, you may indicate that here to bypass the “Comments to the Author” section, enter your conflict of interest statement in the “Confidential to Editor” section, and submit your "Accept" recommendation.

Reviewer #1: All comments have been addressed

Reviewer #2: All comments have been addressed

2. Is the manuscript technically sound, and do the data support the conclusions?

Reviewer #1: Yes

Reviewer #2: Yes

3. Has the statistical analysis been performed appropriately and rigorously? 

Reviewer #1: Yes

Reviewer #2: Yes

4. Have the authors made all data underlying the findings in their manuscript fully available?

Reviewer #1: Yes

Reviewer #2: Yes

5. Is the manuscript presented in an intelligible fashion and written in standard English?

Reviewer #1: Yes

Reviewer #2: Yes

6. Review Comments to the Author

Reviewer #1: Dear Authors,

Thank you for addressing the comments on the manuscript. In this new version, the clarity of the manuscript significantly improved. After reviewing the revised version of the manuscript, I have only a few minor comments.

P.2, l.35-36 – I suggest starting the sentence including the word “Older women…” and change “with age” by “compared to the younger individuals” to make the sentence clearer for the readers.

P.7 Stair descent subsection – It seems that both stair descent and restabilization phases are being described in this subsection. However, there is another subsection (Restabilization) for the restabilization phase description. Please, review.

P.12, l.256-258 – I suggest including in this sentence the information that these data are about age comparisons, because in the way it is written, it seems that the authors are talking about gender differences.

Reviewer #2: The authors addressed all of the comments including major and minor points.

No coroners about dual publication, research ethics, or publication ethics.

7. PLOS authors have the option to publish the peer review history of their article (what does this mean?). If published, this will include your full peer review and any attached files.

Reviewer #1: No

Reviewer #2: No

---

## [Author Response · Author response to Decision Letter 1]

17 Dec 2020

Dear Editor and Reviewers,

we would like to thank for your detailed and useful comments and recommendations. All changes in the manuscript are highlighted. 

Below you will find our point-by-point statements to your comments. Changes are highlighted in green for Editor and in purple for Reviewer. 

Yours sincerely

The authors

EDITOR

1) Please improve the rationale of the abstract and remove it. For example, why are more readily seen differences between age and gender under unstable conditions?

Done. We changed rationale as recommended. 

2) Please double check the writing on line 64 and 222.

Sentences were rewritten.

3) Please change hypothesize to hypothesized. - line 82

Done.

4) I have two possible limitations of the study. Please addressed them. 

I) there are fallers in the older adults sample, but not in young. This should be mentioned.

II) the number of young and older participants (especially for women) is many different. This should be mentioned too.

Done. Mentioned limitations were added.

REVIEWER

P.2, l.35-36 – I suggest starting the sentence including the word “Older women…” and change “with age” by “compared to the younger individuals” to make the sentence clearer for the readers.

Done. Sentence was rewritten as recommended. 

P.7 Stair descent subsection – It seems that both stair descent and restabilization phases are being described in this subsection. However, there is another subsection (Restabilization) for the restabilization phase description. Please, review.

Word "Restabilization" was replaced by "This phase".

P.12, l.256-258 – I suggest including in this sentence the information that these data are about age comparisons, because in the way it is written, it seems that the authors are talking about gender differences.

Done. Sentence was rewritten.

---

## [Editor Report · Decision Letter 2]

21 Dec 2020

Age-related differences in stair descent balance control: are women more prone to falls than men?

PONE-D-20-27933R2

Dear Dr. Kovacikova,

We’re pleased to inform you that your manuscript has been judged scientifically suitable for publication and will be formally accepted for publication once it meets all outstanding technical requirements.

Kind regards,

Fabio A. Barbieri, PhD

Academic Editor

PLOS ONE
---

## [Editor Report · Acceptance letter]

26 Dec 2020

PONE-D-20-27933R2 

Age-related differences in stair descent balance control: are women more prone to falls than men? 

Dear Dr. Kovacikova:

I'm pleased to inform you that your manuscript has been deemed suitable for publication in PLOS ONE. Congratulations! Your manuscript is now with our production department. 

Kind regards, 

on behalf of

Dr. Fabio A. Barbieri 

Academic Editor

PLOS ONE